# Predictors of bleeding complications during catHeter-dirEcted thrombolysis for peripheral arterial occlusions (POCHET)

**Barend M. Mol**[1], **Maarten C. Verwer**[1], **Rob Fijnheer**[2], **Jasper Florie**[3], **Oscar A. Groot**[4], **Falco Hietbrink**[5], **Mathilde Nijkeuter**[6], **Evert-Jan P. A. Vonken**[7], **Vincent van Weel**[8], **Dominique P. V. de Kleijn**[1], **Gert J. de Borst**[1]*, on behalf of the POCHET study group[¶]

1 Department of Vascular Surgery, University Medical Center Utrecht, Utrecht, The Netherlands,
2 Department of Hematology, Meander Medical Center, Amersfoort, The Netherlands, 3 Department of Interventional Radiology, Meander Medical Center, Amersfoort, The Netherlands, 4 Intensive Care Department, Meander Medical Center, Amersfoort, The Netherlands, 5 Department Trauma Surgery, University Medical Center Utrecht, Utrecht, The Netherlands, 6 Department of Vascular Medicine, University Medical Center Utrecht, Utrecht, The Netherlands, 7 Department of Interventional Radiology, University Medical Center Utrecht, Utrecht, The Netherlands, 8 Department of Vascular Surgery, Meander Medical Center, Amersfoort, The Netherlands

¶ Members of the POCHET STUDY Group are provided in the acknowledgments
* G.J.deBorst-2@umcutrecht.nl

**Data Availability Statement:** No datasets were generated or analysed during the current study. All relevant data from this study will be made available upon study completion

## Abstract

### Introduction

The risk of major bleeding complications in catheter directed thrombolysis (CDT) for acute limb ischemia (ALI) remains high, with reported major bleeding complication rates in up to 1 in every 10 treated patients. Fibrinogen was the only predictive marker used for bleeding complications in CDT, despite the lack of high quality evidence to support this. Therefore, recent international guidelines recommend against the use of fibrinogen during CDT. However, no alternative biomarkers exist to effectively predict CDT-related bleeding complications. The aim of the POCHET biobank is to prospectively assess the rate and etiology of bleeding complications during CDT and to provide a biobank of blood samples to investigate potential novel biomarkers to predict bleeding complications during CDT.

### Methods

The POCHET biobank is a multicentre prospective biobank. After informed consent, all consecutive patients with lower extremity ALI eligible for CDT are included. All patients are treated according to a predefined standard operating procedure which is aligned in all participating centres. Baseline and follow-up data are collected. Prior to CDT and subsequently every six hours, venous blood samples are obtained and stored in the biobank for future analyses. The primary outcome is the occurrence of non-access related major bleeding complications, which is assessed by an independent adjudication committee. Secondary outcomes are non-major bleeding complications and other CDT related complications. Proposed biomarkers to be investigated include fibrinogen, to end the debate on its usefulness, anti-plasmin and D-Dimer.

**Funding:** The author(s) received no specific funding for this work.

**Competing interests:** The authors have declared that no competing interests exist.

## Discussion and conclusion

The POCHET biobank provides contemporary data and outcomes of patients during CDT for ALI, coupled with their blood samples taken prior and during CDT. Thereby, the POCHET biobank is a real world monitor on biomarkers during CDT, supporting a broad spectrum of future research for the identification of patients at high risk for bleeding complications during CDT and to identify new biomarkers to enhance safety in CDT treatment.

## Introduction

Acute limb ischemia (ALI) is a potentially lethal condition in which the viability of the limb is threatened due to an acute decrease in limb perfusion [1]. Following the Rutherford classification for ALI [2], catheter directed thrombolysis (CDT) is generally the preferred treatment strategy in viable (Rutherford I) or marginally threatened (Rutherford IIa) ALI [1]. The benefit of CDT should be counterbalanced with the inherent associated risk of bleeding complications which remain a major concern when applying CDT. Bleeding complications rates range from 11% to 56%, with an average complication rate of 18% with almost half of these being major [3]. One of the most severe CDT related major complication is haemorrhagic stroke, which may occur in up to 4% of patients undergoing CDT for ALI [4].

Periprocedural prediction of bleeding complications during CDT has been topic of research since long.

Standard operating procedures (SOP) of CDT for ALI are heterogeneous, which makes it challenging to compare results of previous single centre cohort studies [3,5]. This heterogeneity is not caused by negligence, but by a lack of research on the best treatment standards for CDT, as most SOP are still based on landmark trials like the 'Surgery versus Thrombolysis for Ischemia of the Lower Extremity' (STILE, 1994) trial and the 'Thrombolysis Or Peripheral Arterial Surgery' (TOPAS, 1998) trial [6,7]. These studies have been performed nearly thirty years ago and included patients are likely not representable for the current CDT-population. Duration of ALI for STILE inclusions was up to 6 months and patients enrolled in the TOPAS trial were treated with intravenous heparin up to two times the baseline level of activated partial thromboplastin time (aPTT) plus intra-arterially CDT, causing a high rate major bleeding complications. Despite the heterogeneity in SOP, plasma fibrinogen level (PFL) monitoring has been widely used to predict the occurrence of major bleeding complications, as depicted by a recent survey among physician members of the Society of Interventional Radiology [8]. This practice is mainly based on the observed association between low PFL and the occurrence of major bleeding complications in the STILE trial, although PFL measurement in this study was done only after CDT [6]. Furthermore, a recent systematic review described the predictive value of PFL for major bleeding during CDT and stated that it could not be clarified for which threshold PFL is associated with haemorrhagic complications. Moreover, the role of PFL, in general, as predictor for haemorrhagic complications during CDT remains unproven [9]. Currently, the European Society for Vascular Surgery (ESVS) guidelines on ALI do not recommend routine monitoring of PFL during thrombolysis, mainly due to the lack of high quality evidence underlying its practice [1]. This finding advocates the necessity to prospectively investigate the predictive value of PFL, as well as a need for identification of other potential predictors.

The aim of POCHET biobank (**P**redictors **O**f bleeding **C**omplications during cat**H**eter dir**E**cted **T**hrombolysis for peripheral arterial occlusions) is to prospectively collect data of ALI

patients treated with CDT to assess the rate and etiology of bleeding complications during CDT and to establish a prospective biobank of blood samples taken prior and during CDT to investigate current and potential new biomarkers that predict bleeding complications.

## Methods and design

### Study design

The POCHET biobank is a multicentre prospective biobank study including all consecutive patients eligible for CDT for the treatment of ALI of the lower limb. All patients are treated according to the standard operating procedure for CDT which is aligned in each participating centre. Patients are treated with either Urokinase or Alteplase, as effects and risks are similar [10]. Results will be corrected for thrombolytic medication given. The standard operating procedure is further elaborated in the (S1 File) Written informed consent is collected and included patients donate venous blood samples prior to CDT and simultaneous to standard blood collection every six hours for the duration of thrombolysis. With a maximum CDT-duration of 72 hours, the number of blood collections are maximised at 13 times. As the nature of this study is an observational biobank study, registration as a clinical trial is not applicable [11].

### Ethical approval

This biobank is conducted with approval of the Biobank Research Ethics Committee (BREC) of the University Medical Center (UMC) Utrecht (approval number 19–731, date of approval 30 march 2021). Within this ethical approval, donating a maximum of 136.5 ml blood to the biobank is considered safe and does not affect standard healthcare [12]. For all participating hospitals, a transfer of human material and associated personal data agreement is signed in order to store all collected samples in the central biobank in the UMC Utrecht, the Netherlands. Before plasma samples and its corresponding clinical data can be analysed for a specific research question, approval of the BREC is necessary.

### Inclusion and exclusion criteria

When the treating physician decides CDT is the treatment of choice for ALI, patients must fulfil the following inclusion criteria for the POCHET study:

- Subacute (<6 weeks) thrombotic occlusion of a native artery, bypass or stent in the iliac or femorodistal artery.

- 18 years of age.

- Written informed consent.

### Exclusion criteria

A patient is not able to participate in this biobank study if one of the following criteria is present:

- No treatment indication for CDT

- Not willing to or not able to provide written informed consent

- Absolute or relative contra-indication for thrombolytic therapy (see S1 File)

### Treatment location

In all participating centres, a dedicated angio suite is used for all angiograms made throughout the CDT process: for confirmation of the occlusion, for (mechanical) thrombectomy prior to CDT if deemed necessary and for evaluation of CDT treatment. Patients are being monitored in at least a medical care unit facility during CDT (See S1 File, sub 4, 7 and 9).

### Sample size

The POCHET biobank is an ongoing biobank storing blood samples of patients undergoing thrombolytic therapy for future biomarker research questions. As knowledge develops on predicting biomarkers on bleeding complications during CDT, new research questions are likely to arise constantly. As such, a sample size calculation on a biobank is not possible. The first blood sample assessments on bleeding biomarkers will be done at 100 inclusions. Currently, POCHET biobank has approval for 500 participants, but is likely to expand after the first biomarkers on bleeding complications are explored and higher statistical power might be needed. For this, it is anticipated that the POCHET biobank will become the biggest cohort of plasma samples of patients which are treated with CDT for ALI.

### Endpoint definitions

The primary endpoint is the occurrence of non-acces site related major bleeding during CDT. Major bleeding complication is defined as bleeding related death, intracranial bleeding, intra-organ bleeding, surgical -, radiological—or endoscopic intervention related to bleeding or necessity of inotropic medication.

Secondary endpoints consists of non-major bleeding complications leading to early cessation of thrombolysis, other bleeding complications, and treatment related complications such as trash, compartment syndrome, 30 day re-intervention,30 day major amputation and non-bleeding related death. A list of all endpoint definitions is available in the (S2 File).

An independent clinical adjudication committee is constructed for the assessment of bleeding complications and consists of a trauma surgeon, haematologist and internal medicine physician. The committee will independently determine the presence of bleeding complications, its location and clinical implications with a standardized form and based on anonymized patient data. In this way, all relevant bleeding classifications for thrombolysis can be deducted and evaluated in a blinded fashion on their clinical implications. In case of disagreement between the three members, discrepancies will be discussed until unanimous decision (See Fig 1).

### Lab manual for blood samples

For the duration of CDT, simultaneous to standard procedures, venous blood samples will be collected every 6 hours consisting of an extra 6 ml EDTA and 4.5 ml Citrate tube, starting with the first blood collection before the enhancement of the CDT treatment (See Fig 2). When CDT stops, blood collections for the biobank will stop immediately. Venous blood collection is done by a second peripheral venous catheter which will not be used for fluid or medication administration. Blood samples are not collected by using the sheath or thrombolysis catheter, as thrombolytic medication might interfere with the coagulation biomarkers used for future analysis. Blood samples are centrifuged at 2000g for 10 minutes at room temperature and plasma is stored in 0.5ml tubes within two hours after collection at -80˚ C. These samples are then transported from participating centres to a central biobank of the University Medical

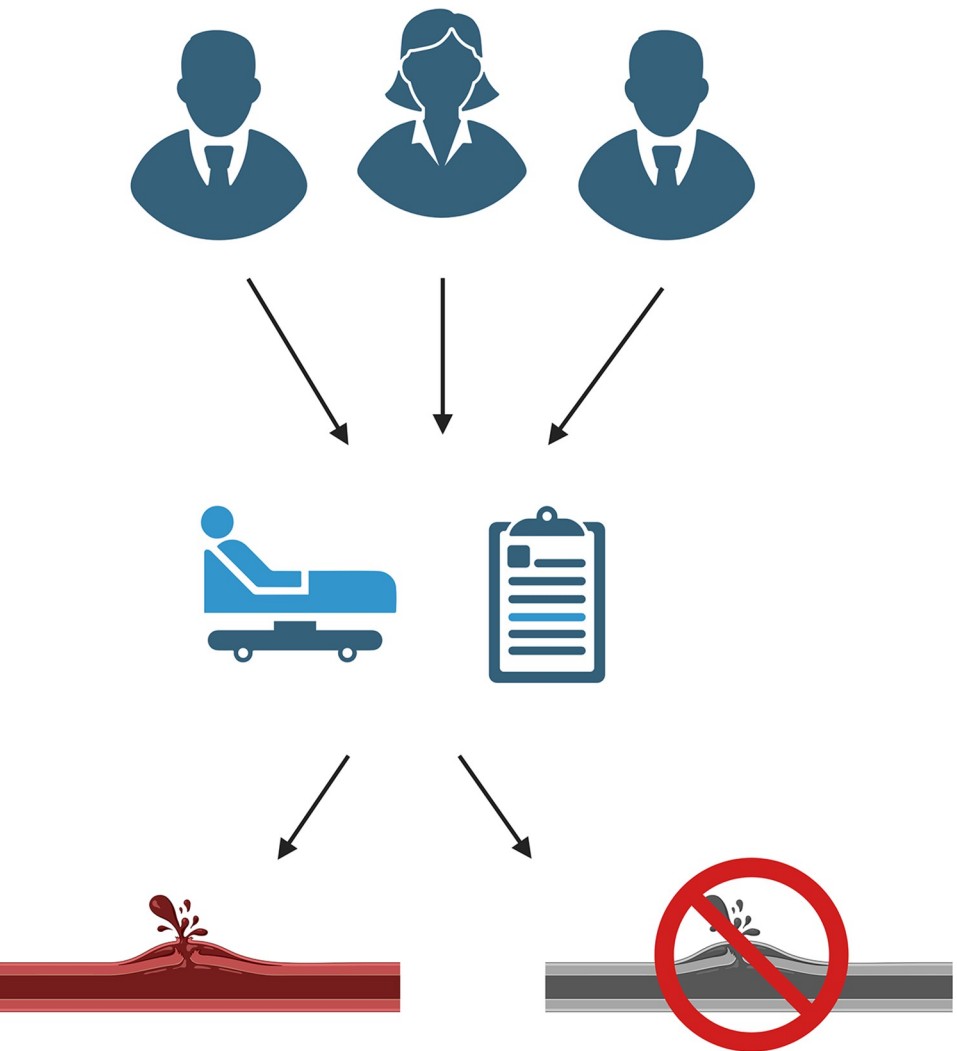

**Fig 1. The independent adjudication committee will define and score presence and severity of bleeding complications in a blinded fashion.**

Center Utrecht and again stored at -80° C awaiting future analysis (See Fig 3). All future analysis will be done in a single laboratory facility,

## Data collection

Data will be prospectively collected from electronic medical records and pseudo-anonymized by the study data manager. Only the data manager will have access to the pseudo-anonymisation key. Data regarding patient characteristics at presentation such as comorbidities, medication use, occlusion specifics and data of hospital admission and follow-up will all be collected and stored in the data management system Castor EDC. Pseudo-anonymized blood samples will be stored and analysed later by trained laboratory personnel. All laboratory outcome data will be digitally stored in a secured location at the University Medical Center Utrecht. The anonymized and encrypted data is only available for authorized research personnel. A list of all included variables is available in the (S3 File). See Fig 4 for a summary of the POCHET biobank.

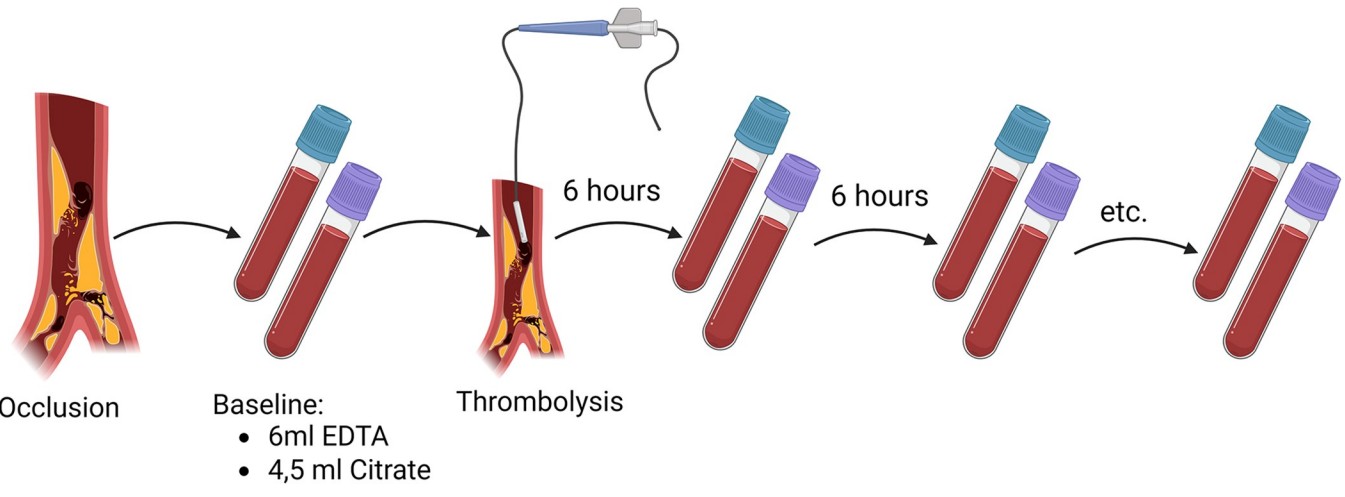

**Fig 2. Blood sample collection protocol.**

## Patient and public involvement

Currently, in the explorative phase of the POCHET biobank we have chosen not to involve patients with the establishment of the biobank. But, when the POCHET biobank has stored blood samples of many different enrolled patients, the database can be used to contribute in a broad spectrum of research questions, for which both patient—or national cardiovascular associations can play a key role in formulating and designing future research.

## Statistical analysis

Baseline characteristics will be presented as mean ± standard deviation in case of normally distributed data and as median ± interquartile range for skewed data. Categorical variables will be stated as absolute numbers and percentages. Groups will be made according to bleeding or no bleeding complication during thrombolytic treatment and baseline characteristics will be compared using T-Tests, Wilcoxon Rank-sum test and Pearson's Chi Square tests when applicable. The predictive value of different haemostatic components on bleeding complications will be analysed by regression analysis. The discriminative power will be measured by C-statistics and Net Reclassification Index. Univariate and Multivariate Cox Regression analysis will be used to test for predictors for bleeding complications, mortality and limb salvage. Proportional hazard assumption will be used to correct for competing interest.

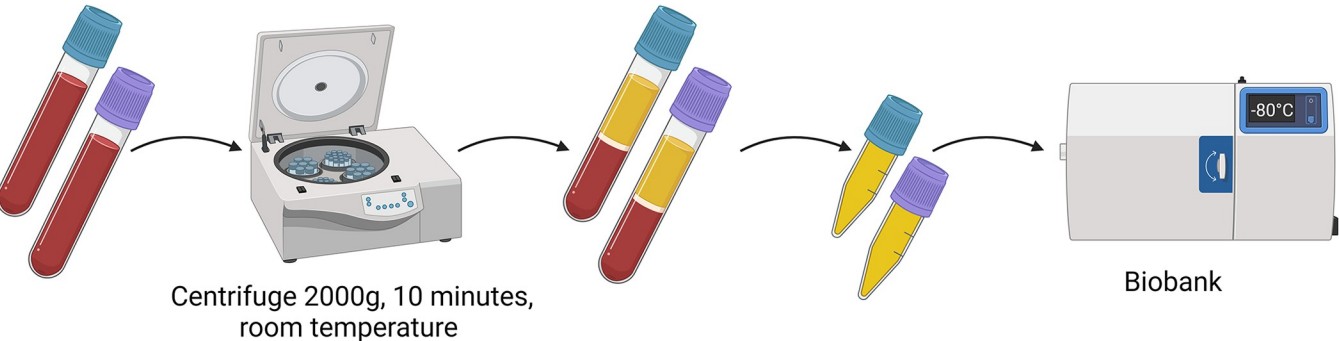

**Fig 3. Processing of blood samples.**

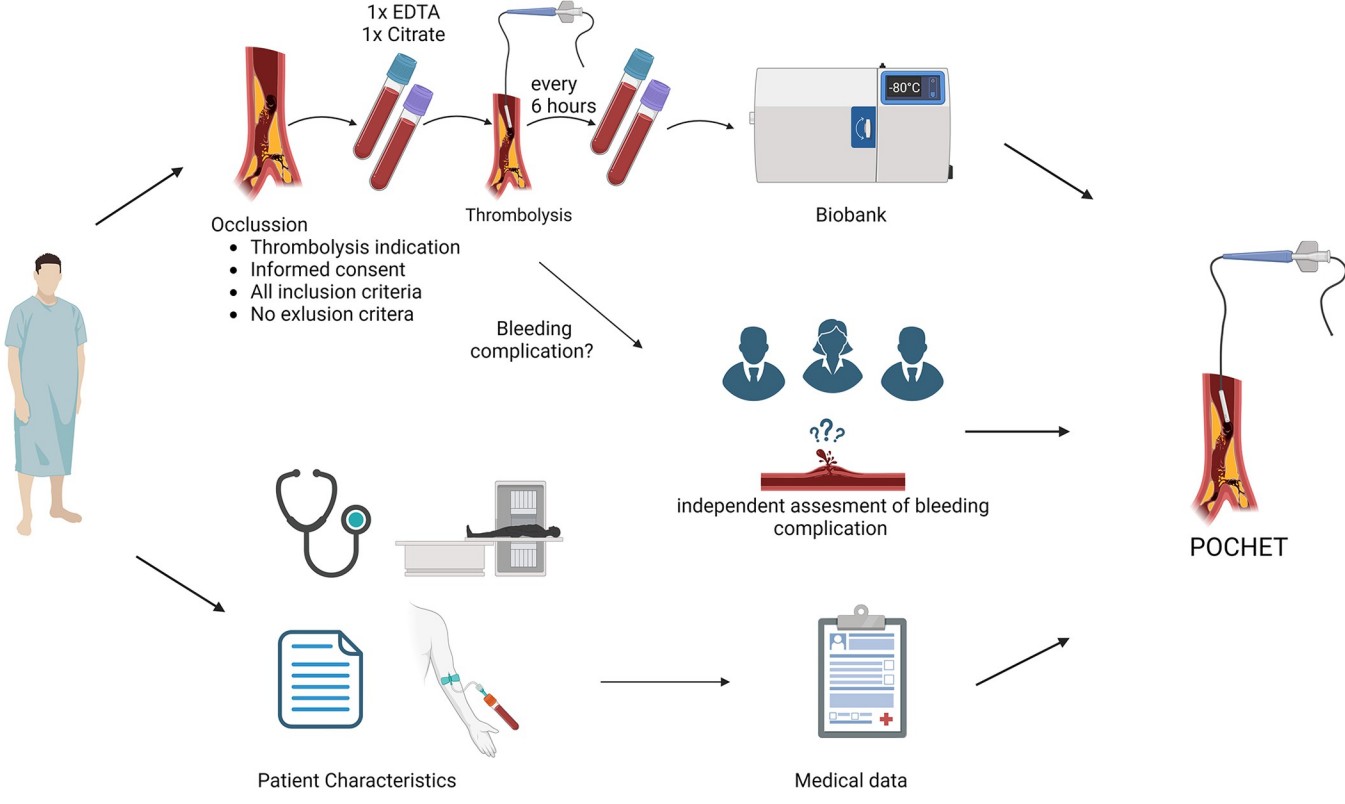

**Fig 4. Summary of POCHET study.**

## Discussion

POCHET is the first large scale prospective biobank including patients with ALI undergoing CDT treatment. By prospectively collecting data from patients and obtaining sequential blood samples, it is possible to provide contemporary characteristics and outcomes of nowadays ALI-patients undergoing CDT and investigate the role of potential (new) biomarkers to predict major bleeding complications during CDT for ALI.

The assessment of new biomarkers to predict bleeding complications requires a uniform definition of a bleeding complication, classification in major or minor and access versus non access related bleeding complications. Although multiple bleeding classifications are available, none are specifically appropriate for CDT in ALI because they are derived from non-surgical cohorts and differ when calling a bleeding complication major [13–16]. The international society of thrombosis and haemostasis (ISTH), for instance, considers a drop of haemoglobin of 1,24 mmol/L (2g/dL) related to haemorrhage a major bleeding, although this might not have any clinical consequence to the CDT therapy [16]. Other bleeding classifications like the Thrombolysis In Myocardial Infarction (TIMI) and Bleeding Academic Research Consortium (BARC) are developed in patients with cardiac diseases and anti-coagulation therapies and thereby not reflecting patients with ALI [13,14]. The Clavien-Dindo classification, although widely used among surgical patients, is a very well defined grading classification for complications in general, but not specific for bleeding complications [17]. Therefore, an independent adjudication committee is implemented to this study, to assess bleeding complications from a basic clinical perspective such as bleeding location, haemoglobin drop, therapy needed and the necessity to seize thrombolytic therapy. In this way, bleeding complications are descriptive

and different existing bleeding classifications can be deducted separately. Furthermore, for the definition of the primary endpoint of this study, non-access site related major bleeding complication, the commonly used criterion "necessity of transfusion of 2 or more units of blood" is intentionally not used as it is highly linked to the type of surgery or intervention in the surgical population [18]. A criterion covering events of excessive and unexpected blood loss are marked with the type of intervention that was implemented. That is why these events were incorporated in the definition of major bleeding complication for the POCHET trial, such as re-intervention. Nevertheless, blood transfusion in relation to any bleeding complication is relevant information and will be collected. With this information, different bleeding classifications can then be analysed for which classification is most clinically relevant for the population of patients receiving CDT.

The POCHET biobank has a broad scope of future research that can be conducted with the available samples and data. With an aligned thrombolytic therapy for every patient, existing risk factors for bleeding complications can be verified. Obviously, the usefulness of fibrinogen catches the eye as one of the first research questions to be answered when trying to end the debate on this matter, as its absolute value or degradation speed remains unproven in the prediction of bleeding complications during CDT [9,19,20]. Furthermore, clinical risk factors for bleeding complications, such as an older age, cardiac history, limb sensibility and/or motor deficit on presentation, platelet count, and occluded synthetic grafts, can be verified with this biobank [21,22].

Several potential biomarkers could be hold promise as a predictor for major bleeding during CDT. For instance Von Willebrand Factor (VWF), the platelet adhesion protein. VWF-platelet complexes are heavily degraded in tissue plasminogen activator therapy, such as alteplase, leading to fibrinolysis [23]. The subsequent inability for thrombus formation by plasminogen activation elsewhere can thereby lead to bleeding complications [24].

Furthermore, anti-plasmin and plasminogen activator inhibitor both serve as the main down regulators in fibrinolysis. These regulators must be surmounted during thrombolytic therapy around the thrombus for effective lysis, but still be effective in systemic circulation to avoid bleeding [25]. Low baseline levels before CDT or degradation of these down regulators during treatment might lead to bleeding complications.

Another promising marker is D-dimer, a major fibrin degradation product that is released upon cleavage of crosslinked fibrin by plasmin. Widely used in the screening for embolic disease such as deep venous thrombosis, a recent review showed potential prognostic value for D-dimer on safety and outcome in thrombolysis [26].

A recent addition in biomarkers are the proteins in plasma extra-cellular vesicles (EVs), bilayer lipid membrane particles excreted by each cell, containing important information concerning coagulation [27]. For instance, platelet-derived and red blood cell-derived EVs both support factor XII dependent thrombin generation and thereby enhancing the coagulation cascade [28]. Degradation of these vesicles during CDT might impede coagulation and thereby increase the risk of bleeding.

Furthermore, fundamental human research on coagulation and fibrinolysis is possible with the POCHET biobank, with both samples within one individual patient prior to thrombolysis and samples on which the homeostasis is completely shifted towards anti-coagulation. For instance, little is known about the effect of lipoprotein Lp(a), a risk factor for atherosclerotic disease, and induced thrombolysis. One effect might be that Lp(a) may compete with plasminogen and thereby increase thrombolysis time [29]. With increased thrombolysis time with high levels of Lp(a), bleeding complications might be more frequent in these patients and merits further analysis.

In summary, POCHET will be the first large scale biobank collecting consecutive blood samples of patients with ALI that undergo CDT. Related to its design, POCHET will 1) help characterise the consecutive ALI population treated with CDT, 2) help identify novel biomarkers to predict major bleeding complications and hence aid physicians and research scientists to create a safer thrombolytic therapy and 3) support a wide range of diverse research projects in the field of coagulation and related interventions.

## Supporting information

**S1 File. Standard thrombolytic treatment of the lower limb.**
(DOCX)

**S2 File. Endpoint definitions of the POCHET study.**
(DOCX)

**S3 File. All variables included in the POCHET study.**
(DOCX)

## Acknowledgments

Members of the POCHET study group:

G. J. de Borst Department of Vascular Surgery, University Medical Center Utrecht, Utrecht, The Netherlands

M.K. Dinkelman Department of Vascular Surgery, Elisabeth TweeSteden Hospital, Tilburg, The Netherlands

D.E.J.G.J. Dolmans Department of Vascular Surgery, Diakonessenhuis Hospital, Utrecht, The Netherlands

W.M. de Fijter Department of Vascular Surgery, Elisabeth TweeSteden Hospital, Tilburg, The Netherlands

D.P.V. de Kleijn Department of Vascular Surgery, University Medical Center Utrecht, Utrecht, The Netherlands

E.S. van Hattum Department of Vascular Surgery, University Medical Center Utrecht, Utrecht, The Netherlands

C.E.V.B. Hazenberg Department of Vascular Surgery, University Medical Center Utrecht, Utrecht, The Netherlands

J.A. van Herwaarden Department of Vascular Surgery, University Medical Center Utrecht, Utrecht, The Netherlands

J.M.M. Heyligers Department of Vascular Surgery, Elisabeth TweeSteden Hospital, Tilburg, The Netherlands

A. Jansze Department of Vascular Surgery, University Medical Center Utrecht, Utrecht, The Netherlands

M.C. Loubert Department of Vascular Surgery, Meander Medical Center, Amersfoort, The Netherlands

B.M. Mol Department of Vascular Surgery, University Medical Center Utrecht, Utrecht, The Netherlands

S.K. Nagesser Department of Vascular Surgery, Diakonessenhuis Hospital, Utrecht, The Netherlands

B.J. Petri Department of Vascular Surgery, University Medical Center Utrecht, Utrecht, The Netherlands

M. Teraa Department of Vascular Surgery, University Medical Center Utrecht, Utrecht, The Netherlands

R.J. Toorop Department of Vascular Surgery, University Medical Center Utrecht, Utrecht, The Netherlands

W.J. Thijsse Vascular Surgery, Meander Medical Center, Amersfoort, The Netherlands

P.W.H.E. Vriens Department of Vascular Surgery, Elisabeth TweeSteden Hospital, Tilburg, The Netherlands

M. de Vries Department of Vascular Surgery, Diakonessenhuis Hospital, Utrecht, The Netherlands

V. van Weel Vascular Surgery, Meander Medical Center, Amersfoort, The Netherlands

Lead author of the POCHET Study Group: G.J. de Borst. E-mail address: G.J.deBorst-2@umcutrecht.nl

All figures were created with www.biorender.com.

## Author Contributions

**Conceptualization:** Maarten C. Verwer, Rob Fijnheer, Jasper Florie, Oscar A. Groot, Vincent van Weel, Dominique P. V. de Kleijn, Gert J. de Borst.

**Methodology:** Barend M. Mol, Maarten C. Verwer, Rob Fijnheer, Jasper Florie, Oscar A. Groot, Evert-Jan P. A. Vonken, Vincent van Weel, Dominique P. V. de Kleijn, Gert J. de Borst.

**Project administration:** Barend M. Mol.

**Supervision:** Dominique P. V. de Kleijn, Gert J. de Borst.

**Validation:** Falco Hietbrink, Mathilde Nijkeuter.

**Writing – original draft:** Barend M. Mol, Maarten C. Verwer.

**Writing – review & editing:** Rob Fijnheer, Jasper Florie, Oscar A. Groot, Falco Hietbrink, Mathilde Nijkeuter, Evert-Jan P. A. Vonken, Vincent van Weel, Dominique P. V. de Kleijn, Gert J. de Borst.

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
