## [Decision Letter · Decision Letter 0]

23 Jan 2024

PONE-D-23-42662Predictors Of bleeding Complications during catHeter-dirEcted Thrombolysis for peripheral arterial occlusions (POCHET)PLOS ONE

Dear Dr. Mol,

Thank you for submitting your manuscript to PLOS ONE. After careful consideration, we feel that it has merit but does not fully meet PLOS ONE’s publication criteria as it currently stands. Therefore, we invite you to submit a revised version of the manuscript that addresses the points raised during the review process.

We look forward to receiving your revised manuscript.

Kind regards,

Eyüp Serhat Çalık

Academic Editor

PLOS ONE

Journal Requirements:

2. Please include “Protocol” in the manuscript  title.

3. Please include a copy of the ethics committee approval, which should include the date of approval, the approval number, and signature or stamp from the ethics committee, and English translation, as an "Other" file.

In the Methods section of your revised manuscript, please include the full name of the institutional review board or ethics committee that approved the protocol, the approval or permit number that was issued, and the date that approval was granted.

4. One of the noted authors is a group or consortium: The POCHET study group

In addition to naming the author group, please list the individual authors and affiliations within this group in the acknowledgments section of your manuscript. Please also indicate clearly a lead author for this group along with a contact email address.

**Additional Editor Comments:**

I read the manuscript on this important topic with interest. I think that the study protocol is generally well written and will provide important contributions to those interested in the subject when the study is completed. The manuscript has been reviewed by three reviewers and their concerns are as follows. Please edit your manuscript according to the reviewers' suggestions and provide point-by-point responses. In particular, please provide the ethics committee approval of the study and the clinical trials register.

Reviewers' comments:

Reviewer's Responses to Questions

**Comments to the Author**

1. Does the manuscript provide a valid rationale for the proposed study, with clearly identified and justified research questions?

Reviewer #1: No

Reviewer #2: Yes

Reviewer #3: Yes

2. Is the protocol technically sound and planned in a manner that will lead to a meaningful outcome and allow testing the stated hypotheses?

Reviewer #1: No

Reviewer #2: Yes

Reviewer #3: Yes

3. Is the methodology feasible and described in sufficient detail to allow the work to be replicable?

Reviewer #1: No

Reviewer #2: Yes

Reviewer #3: Yes

4. Have the authors described where all data underlying the findings will be made available when the study is complete?

Reviewer #1: No

Reviewer #2: Yes

Reviewer #3: No

5. Is the manuscript presented in an intelligible fashion and written in standard English?

Reviewer #1: Yes

Reviewer #2: Yes

Reviewer #3: Yes

6. Review Comments to the Author

You may also provide optional suggestions and comments to authors that they might find helpful in planning their study.

Reviewer #1: Overall evaluation: The order of work in this study protocol should be reversed. First full ethical approval, then Clinical Trials registration, then submission of study protocol including statements of full ethical approval and Clinical trial registration number.

It should be clear from the protocol that some data will be collected and interpreted as part of the emergency intervention for proper patient care while some blood samples will be collected for “the future”.

One large problem you will encounter is that most bleeding events, also major, is related to access-site related bleeding that has nothing to do with anticoagulation, coagulation or fibrinolytic disturbances. Have you thought of that? Your primary endpoint should be major non-introducer-related major bleeding, and your estimated sample size should be based on assumption upon this endpoint, and not all major bleeding complications.

Specific comments:

1. Abstract. Add how many centers.

2. Add Clinical Trials registration number

3. Abstract – define major bleeding already here

4. The full ethical approval of this study is missing and should be included before the study protocol is submitted to a journal. In this ethical approval, the maximal amount of extra blood taken from the patient outside of clinical routine should be stated, and a reference given that it is ok to withdraw a certain amount of blood without any clinical consequences.

5. Some kind of power calculation would benefit this study protocol. How many patients would be needed to have some clue whether fibrinogen is a marker of major bleeding?

6. Data collection. The clinical biochemistry assays for example for fibrinogen may differ across centers. Have you checked that? How do you intend to calibrate between assays?

7. Data collection. No use of for instance REDCap data management system?

8. Definition of major bleeding – you should add necessity of transfusion of at least 2 units of blood during the in-hospital stay. Give reference: Schulman et al. J Thromb Haemostat 2010; 8: 202-204. Even though you argue against this, it is clinically relevant since many patients with ALI also have concomitant anemia at admission (around 30%).

9. Statistics. Can you expand on how you will be able to correct for competing interest? I mean if a patient dies, there will be no possibility of major amputation.

10. Limb salvage. Do you mean major amputation? Please define.

11. Ref 12 and 15 are the same.

Reviewer #2: The present study protocol aims to investigate the possible biomarkers to identify patients with high bleeding risk associated with catheter directed thrombolysis. The topic is clinically relevant although the multiple treatment protocols add to the challenge of clinical trial. The use of either Alteplase or Urokinase would make study more standardized. However if target 500 patients will be reached the post hoc comparation between regimen and biomarkers might be possible.

Only some minor notes on the manuscript:

Since present study is a clinical trial it would be logical to be registered as clinical trial at for example web registry ClinicalTrials.gov and registry number to be added to the manuscript.

Study design it would be more informative to add a nice flow chart. The reference to S1 is ok, but nice flow chart would make manuscript more interesting. Also figures 1-4 are nice but flow chart would be more accurate and present figures could support the flow chart?

Allergy against Alteplase or Urokinase not contraindication according S1? Please check absolute and relative contraindications.

Figure 3 one might get impression that the blood samples are collected from introduction sheet close to thrombus, which might even be interesting addition considering the aim of the study?

Biobank approval number available? If it would be nice to add it to the text.

Reviewer #3: Dear authors

Thanks for sharing your protocol. I have some few suggestions/doubts that may improve your data.

1)Title and abstract are interesting and highlight main points of the study. I think abstract must draw attention to the readers. I suggest report which biomarkers will be included at your research at the abstract.

2) Inclusion criteria: How severe were the included patients related to the limb perfusion? Only ALI Rutherford I and IIa patients were included? Please, answer - You must be very clear as this is a very crucial feature to take in accout. Explain

3) Exlusion criteria: IIB and II Rutherford ALI categories. Yes or not? Please, explain

4) About potential biomarkers: I suggest you describe how they will be measured at blood samples - your protocol and findings must be reproductible for other groups , mainly anti-plasmin, plasminogen activator inhibitor and VES.

5) Will fibinolytic agent be standartized? Is alteplase? Or other agents will be allowed? Please , be as clearer as possible.

6) Both ischemic upper and lower limbs will be included? Pleases, specify it.

7) I suggest define 02(two) groups of endopoints: efficacy endpoints and safety endpoints (besides bleeding complications, I suggest reporting "limb salvage"). All your variables MUST be reported.

8) Is your protocol registered (or being registered) on a platform like clinical trials.gov?

Please pay attention, mainly at question 2. It is very important to define your inclusion (and exclusion) criteria - it may be a potential source of bias

7. PLOS authors have the option to publish the peer review history of their article (what does this mean?). If published, this will include your full peer review and any attached files.

Reviewer #1: No

Reviewer #2: No

Reviewer #3: No

---

## [Author Response · Author response to Decision Letter 0]

28 Mar 2024

Response to Reviewers 

Journal Requirements:

1. Please ensure that your manuscript meets PLOS ONE's style requirements, including those for file naming. The PLOS ONE style templates can be found at . 

 The manuscript has been revised to the PLOS ONE’s style requirements

2. Please include “Protocol” in the manuscript title.

 “Protocol” has been added to the manuscript title 

3. Please include a copy of the ethics committee approval, which should include the date of approval, the approval number, and signature or stamp from the ethics committee, and English translation, as an "Other" file.

In the Methods section of your revised manuscript, please include the full name of the institutional review board or ethics committee that approved the protocol, the approval or permit number that was issued, and the date that approval was granted.

 The original letter of approval of the Biobank Research Ethics Committee and it’s English translation has been included as an “other” file with the latest submission. 

 Under the “ethical approval” header within the “Methods and Design” section, the approval number and the date of approval is added. 

4. One of the noted authors is a group or consortium: The POCHET study group

In addition to naming the author group, please list the individual authors and affiliations within this group in the acknowledgments section of your manuscript. Please also indicate clearly a lead author for this group along with a contact email address.

 The affiliation of the members of the POCHET study group have been added in the acknowledgment section. Also the lead author of the POCHET study group is indicated with a contact email address. 

Additional Editor Comments:

I read the manuscript on this important topic with interest. I think that the study protocol is generally well written and will provide important contributions to those interested in the subject when the study is completed. The manuscript has been reviewed by three reviewers and their concerns are as follows. Please edit your manuscript according to the reviewers' suggestions and provide point-by-point responses. In particular, please provide the ethics committee approval of the study and the clinical trials register.

Reviewers' comments:

Reviewer's Responses to Questions

Comments to the Author

1. Does the manuscript provide a valid rationale for the proposed study, with clearly identified and justified research questions?

Reviewer #1: No

Reviewer #2: Yes

Reviewer #3: Yes

2. Is the protocol technically sound and planned in a manner that will lead to a meaningful outcome and allow testing the stated hypotheses?

Reviewer #1: No

Reviewer #2: Yes

Reviewer #3: Yes

3. Is the methodology feasible and described in sufficient detail to allow the work to be replicable?

Reviewer #1: No

Reviewer #2: Yes

Reviewer #3: Yes

4. Have the authors described where all data underlying the findings will be made available when the study is complete?

Reviewer #1: No

Reviewer #2: Yes

Reviewer #3: No

5. Is the manuscript presented in an intelligible fashion and written in standard English?

Reviewer #1: Yes

Reviewer #2: Yes

Reviewer #3: Yes

6. Review Comments to the Author

You may also provide optional suggestions and comments to authors that they might find helpful in planning their study.

Reviewer #1: Overall evaluation: The order of work in this study protocol should be reversed. First full ethical approval, then Clinical Trials registration, then submission of study protocol including statements of full ethical approval and Clinical trial registration number.

It should be clear from the protocol that some data will be collected and interpreted as part of the emergency intervention for proper patient care while some blood samples will be collected for “the future”.

One large problem you will encounter is that most bleeding events, also major, is related to access-site related bleeding that has nothing to do with anticoagulation, coagulation or fibrinolytic disturbances. Have you thought of that? Your primary endpoint should be major non-introducer-related major bleeding, and your estimated sample size should be based on assumption upon this endpoint, and not all major bleeding complications.

Dear reviewer,

The authors would like to thank you for your elaborate evaluation of the study and your very thorough comments. We adjusted the manuscript on these topics and clarify why we made certain decisions. Before we answer the list below, let us please answer your general comments. 

• The nature of this study is observational and does not assign patients to one or different interventions which are compared between groups. For this study, all included patients are treated according to the predefined and standardized treatment protocol. The assessment of the best type of thrombolysis protocol falls beyond the scope of this study. Also because there are many variables to be studied to define the optimal protocol i.e. type of thrombolysis; dosage; access routing; duration of therapy, additional heparinization; interval of control angio etc etc.

• Donated blood samples are stored in the biobank for future analysis. Based on these future analysis, a new study can be conducted which can analyse possible new biomarkers that predict bleeding complications during thrombolysis. Because of the observational setting of the study, our study is not a clinical trial as defined by the International Committee of Medical Journal Editors (ICMJE) as there is no intervention. The definitions states:

“The ICMJE defines a clinical trial as any research project that prospectively assigns people or a group of people to an intervention, with or without concurrent comparison or control groups, to study the relationship between a health-related intervention and a health outcome”

https://www.icmje.org/recommendations/browse/publishing-and-editorial-issues/clinical-trial-registration.html

For this, the authors believe that a registration as a clinical trial is not applicable. To clarify this, we added “ As the nature of this study is an observational biobank study, registration as a clinical trial is not applicable“ to the study design section in line 120-121 ( the clean revised manuscript. 

• Reviewer #1 stated that the order of work in this study protocol should be reversed, but since the registration as a clinical trial is not applicable, it is of our believe that the suggested order is no longer applicable. To clarify this, we added “ As the nature of this study is an observational biobank study, registration as a clinical trial is not applicable“ to the study design section in line 120-121 of the clean revised manuscript. 

• The suggestion about the major bleeding complications definition is a very important recommendation. There is a very significant difference between access site related bleedings and any other bleeding during catheter direct thrombolysis. This difference was not clearly stated in the manuscript and therefore, the definition of the primary endpoint is edited to “non- access site related major bleeding” in line 165 (clean revised manuscript) in the endpoint definition section and in line 40 (clean revised manuscript) of the abstract. 

Specific comments:

1. Abstract. Add how many centers. We thank the reviewer for this important comment because the number of including centers is important information. The reason we have not included the number of including centers is because the possibility for other centers to join later on in including patients for the POCHET trial. By stating the number of including centers at this point in time, this information will be incorrect when other authors will reference to this protocol paper. It is of course very important information to state the number of including centers when the results of the POCHET biobank are published. Currently, we have two centers including patients and expect more centers to include patients in the year 2024. 

2. Add Clinical Trials registration number. See first bullet point above.

3. Abstract – define major bleeding already here. As mentioned above, the primary endpoint has been changed in the abstract as ‘non-access related major bleeding” as mentioned above. 

4. The full ethical approval of this study is missing and should be included before the study protocol is submitted to a journal. In this ethical approval, the maximal amount of extra blood taken from the patient outside of clinical routine should be stated, and a reference given that it is ok to withdraw a certain amount of blood without any clinical consequences. The comment is on the ethical approval is a very significant recommendation. This study was approved by the Biobank Research Ethics Committee of the University Medical Center Utrecht, which is the central organ for reviewing new biobank protocols. It’s approval letter of the biobank protocol, including English translation, is submitted with this revised manuscript. In this biobank protocol it is stated that during thrombolysis, along with general blood collections every 6 hours, blood collections for the biobank were performed, consisting of a 6 ml EDTA and a 4,5ml citrate tube. With a maximum duration of thrombolysis of 72 hours and an extra donation to the biobank before thrombolysis would start, the maximum blood collections for the biobank per patient will add up to a maximum of 13 separate donations of 10,5 ml per donation. Blood donations to the biobank would stop when thrombolysis stops, which means that in general practice, a lot of patients will donate less than 13 times. This is further clarified in the Method section under sub header “Study design” and “lab manual for blood samples” in lines 122 (clean revised manuscript) and 198 (clean revised manuscript) and stated below. 

Line 122: “With a maximum CDT-duration of 72 hours, the number of blood collections are maximised at 13 times”

Line 198: “When CDT stops, blood collections for the biobank will stop immediately.”

A total maximum of 136,5 ml per patient is taken. Amrein et al (doi 10.1016/j.blre.2011.09.003) reviewed the risks of blood donation, which were mainly vena puncture related or had to do with vasovagal collapses. For this study, these risks are not applicable, as blood samples are collected by a venous catheter. Long term risks of blood donation would mainly be iron deficiency syndrome, which was more seen in people who donated blood more frequently. The patients in our study, only donate blood to the biobank during thrombolysis treatment and the total amount of blood taken is only one third when compared to a regular blood donation (max 136,5ml vs 500 ml). This contributes to the safety of the blood donation for the POCHET biobank, which is very important. This is why it is further elaborated in the manuscript (line 126 of clean revised manuscript, “donating a maximum of 136.5 ml blood to the biobank is considered safe and does not affect standard healthcare”.) and in this letter.

5. Some kind of power calculation would benefit this study protocol. How many patients would be needed to have some clue whether fibrinogen is a marker of major bleeding?

Thank you for this important recommendation. A power calculation in clinical research aids researchers in the risk of under or over inclusion. However, the POCHET study is not a clinical study, but a biobank. As previous studies mostly had a retrospective analysis as design, there is a broad range of expected endpoints, which also differed in their definitions amongst studies. As such, the prospective POCHET registry using strict defined endpoints and standardized protocol will help to set the new outcome parameters which can subsequently be used to perform power calculations for future thrombolysis based studies.

The most important value of a biobank is that it holds tissue or blood for future research for all kinds of research questions. Every research question can have its own effect and thereby have its own power calculation. This is further elaborated in the subheader “Sample size” in the “Methods and design” section. 

6. Data collection. The clinical biochemistry assays for example for fibrinogen may differ across centers. Have you checked that? How do you intend to calibrate between assays?

This is a very important remark. All plasma samples will be stored temporarily in the participating centers at -80° Celsius. No biochemistry assay is being done in different centers. Samples are transported on dry ice every few months to the central biobank in Utrecht where they again will be stored at -80° Celsius. All biochemistry assays will then be done in a single laboratory. This is further elaborated in the subheader “Lab manual for blood samples” in the “methods and design “ section in line 194 of the clean revised manuscript and stated below.

“All future analysis will be done in a single laboratory facility,” 

7. Data collection. No use of for instance REDCap data management system?

Thank you for this comment. Although it is stated that the data is securely stored, the way it is stored is not mentioned in the manuscript. This is changed in the revised manuscript in line 203 of the clean revised manuscript and stated below

“Data regarding patient characteristics at presentation such as comorbidities, medication use, occlusion specifics and data of hospital admission and follow-up will all be c

---

## [Editor Report · Decision Letter 1]

15 Apr 2024

Predictors Of bleeding Complications during catHeter-dirEcted Thrombolysis for peripheral arterial occlusions (POCHET)

PONE-D-23-42662R1

Dear Dr. Mol,

We’re pleased to inform you that your manuscript has been judged scientifically suitable for publication and will be formally accepted for publication once it meets all outstanding technical requirements.

Kind regards,

Eyüp Serhat Çalık

Academic Editor

PLOS ONE
---

## [Editor Report · Acceptance letter]

26 Apr 2024

PONE-D-23-42662R1 

PLOS ONE

Dear Dr. Mol, 

I'm pleased to inform you that your manuscript has been deemed suitable for publication in PLOS ONE. Congratulations! Your manuscript is now being handed over to our production team.

Kind regards, 

on behalf of

Dr. Eyüp Serhat Çalık 

Academic Editor

PLOS ONE